# QTL Mapping for Resistance to Bacterial Wilt Caused by Two Isolates of *Ralstonia solanacearum* in Chili Pepper (*Capsicum annuum* L.)

**DOI:** 10.3390/plants11121551

**Published:** 2022-06-10

**Authors:** Saeyoung Lee, Nidhi Chakma, Sunjeong Joung, Je Min Lee, Jundae Lee

**Affiliations:** 1Department of Horticulture, Institute of Agricultural Science & Technology, Jeonbuk National University, Jeonju 54896, Korea; lsy4674@hanmail.net (S.L.); nidhi.chakma@griffithuni.edu.au (N.C.); sunjeong7571@gmail.com (S.J.); 2Department of Horticultural Science, College of Agriculture and Life Sciences, Kyungpook National University, Daegu 41566, Korea; jemin@knu.ac.kr

**Keywords:** bacterial wilt, chili pepper, genotyping-by-sequencing, high-resolution melting, quantitative trait loci, single nucleotide polymorphism

## Abstract

Bacterial wilt caused by the β-proteobacterium *Ralstonia solanacearum* is one of the most destructive soil-borne pathogens in peppers (*Capsicum annuum* L.) worldwide. Cultivated pepper fields in Korea face a continuous spread of this pathogen due to global warming. The most efficient and sustainable strategy for controlling bacterial wilt is to develop resistant pepper varieties. Resistance, which is quantitatively inherited, occurs differentially depending on *R. solanacearum* isolates. Therefore, in this study, we aimed to identify resistance quantitative trait loci (QTLs) in two F_2_ populations derived from self-pollination of a highly resistant pepper cultivar ‘Konesian hot’ using a moderately pathogenic ‘HS’ isolate and a highly pathogenic ‘HWA’ isolate of *R. solanacearum* for inoculation, via genotyping-by-sequencing analysis. QTL analysis revealed five QTLs, *Bwr6w-7.2*, *Bwr6w-8.1*, *Bwr6w-9.1*, *Bwr6w-9.2*, and *Bwr6w-10.1*, conferring resistance to the ‘HS’ isolate with *R^2^* values of 13.05, 12.67, 15.07, 10.46, and 9.69%, respectively, and three QTLs, *Bwr6w-5.1*, *Bwr6w-6.1*, and *Bwr6w-7**.1*, resistant to the ‘HWA’ isolate with phenotypic variances of 19.67, 16.50, and 12.56%, respectively. Additionally, six high-resolution melting (HRM) markers closely linked to the QTLs were developed. In all the markers, the mean disease index of the paternal genotype was significantly lower than that of the maternal genotype. The QTLs and HRM markers are expected to be useful for the development of pepper varieties with high resistance to bacterial wilt.

## 1. Introduction

The pepper *Capsicum annuum* L., which belongs to the Solanaceae family, is an important vegetable crop worldwide, with 68% of the total production occurring in Asia [1]. Among Asian countries, South Korea is one of the countries with the highest daily consumption of peppers per person. According to a report from 2019, domestic pepper production and cultivation areas are gradually decreasing [2]. Pepper production and cultivation decline in South Korea is caused by increased volumes of pepper powder import, aging of farmers, low mechanization rate, and various diseases due to continuous cropping and abnormal temperature [3,4]. In particular, diseases caused by plant pathogens are the main causes of decreased crop productivity [5], with 68 species of pepper pathogens reported in South Korea [6]. Among these, bacterial wilt occurs frequently in hot and humid conditions, and the damage to each plant is increasing [4,7,8].

Bacterial wilt occurs extensively in tropical, subtropical, and temperate regions, and can infect more than 50 families of 450 plant species, including solanaceous crops such as pepper, tomato, potato, and tobacco [9]. Bacterial wilt is caused by the soil-borne pathogen *Ralstonia solanacearum*, which belongs to β-Proteobacteria [10,11]. *Ralstonia solanacearum* is capable of long-term overwintering in soil, moves mainly with irrigation water [12], and remains alive in contaminated soil, host plants, and diseased plant residues [12,13]. *Ralstonia solanacearum* moves to the xylem after invading the plant through wounds of the plant’s lateral root and openings in the root cap [14,15]. After successful invasion and colonization of the host, it produces a large amount of exopolysaccharide that interferes with water transport and blocks xylem, leading to the ultimate death of the host plant [16,17,18].

The major virulence factors produced by *R. solanacearum* are exopolysaccharides and cell-wall-degrading enzymes [19,20]. Moreover, the type III secretion system and large repertoire of effectors, which promote bacterial infection, are also the major pathogen determinants of *R. solanacearum* [21,22,23,24]. *Ralstonia solanacearum* is not only less homologous and complex among isolates, but also has a great deal of difference among groups of isolates, as represented by the ‘*Ralstonia solanacearum* species complex (RSSC)’ [25,26,27]. The RSSC is composed of phylotypes, sequevars, and clades based on a new hierarchical classification [26,28]. *Ralstonia solanacearum* is largely classified into five races based on the host range and five biovar types based on physiological or biochemical differences, among which race 1 is the most frequently occurring race in solanaceous crops, with a very broad host range [9,29,30]. Recently, *R. solanacearum* was classified into four phylotypes (I, II, III, and IV) based on ancestral relationship and geographic origin and further as sequevars based on partial analysis and phylogenetic analysis of a gene *egl* encoding the endoglucanase enzyme [26,31,32]. It was also subdivided into eight clades with distinct evolutionary patterns depending on a specific host or climatic conditions [33,34]. *Ralstonia solanacearum* strains isolated from chili peppers in South Korea were classified into race 1 and biovar 3 or 4, of which biovar 4 was reported as the most dominant strain [4,35]. A commercial F_1_ pepper variety ‘Konesian Hot’ has shown resistance to five strains of *R. solanacearum* (groups II to VI) but moderate resistance to the highly pathogenic strain group (group I), which belongs to race 1, biovar 3 or 4 [4].

In the pepper, several accessions including MC4, MC5, HC10, PBC066, PBC631, PI322726, PI322727, PI322728, PI358812, PI369994, PI369998, and PI377688 have been reported to be resistant to bacterial wilt [36,37]. Various strategies were devised for management of bacterial wilt, including weeding, soil disinfection, microbial control, and biological control, although it is not easy to completely prevent bacterial wilt with any of these methods [38,39,40]. Therefore, the most efficient and sustainable strategy to control bacterial wilt disease is the development of resistant varieties [39,41].

Disease resistance can be divided into qualitative and quantitative traits [5]. The use of major *R* (resistance) genes or resistance QTL (quantitative trait loci) pyramiding is an effective strategy to develop resistant varieties [5]. In addition, as the sole dependence on phenotype selection is unreliable, disease resistance breeding complemented by the use of DNA markers, known as marker-assisted selection (MAS), could substantially enhance the selection efficiency and reduce the period of breeding cycles [42,43,44].

In breeding pepper resistance against *R. solanacearum*, the major challenge is the selection of resistant individuals, as the resistance is controlled quantitatively [45]. Therefore, the identification of QTLs responsible for the resistance to bacterial wilt disease is believed to easily and accurately select resistant plants. The QTL *Bw1* conferring resistance to bacterial wilt flanked by one simple sequence repeat (SSR) marker, CAMS451, was identified on chromosome 1 of a pepper linkage map, which was constructed using 106 SSRs and 203 amplified-fragment-length polymorphism (AFLP) markers [44]. The QTL *qRRs-10.1* conferring the resistance flanked by two indel (insertion–deletion) markers, ID10-194305124 and ID10-196208712, was found on pepper chromosome 10 through specific-locus amplified-fragment sequencing combined with bulked segregant analysis [46]. A recent QTL study revealed one major QTL, *pBWR-1*, showing 20.13 to 25.15% of *R^2^* values, on chromosome 1 of a pepper genetic map, which was made using 1550 SNPs from genotyping-by-sequencing (GBS) analysis [47].

The advancement of next-generation sequencing (NGS) technologies has led to accelerating the identification of single-nucleotide polymorphisms (SNPs) and indels, which have the most frequent occurrence in plant genomes [48]. SNP markers, which are based on a difference of one nucleotide between individuals, can be easily applied for a high-throughput screening system that could be used to construct genetic linkage maps and detect nucleotide polymorphisms between alleles [48]. GBS is one of the NGS-based SNP detection technologies that use restriction enzymes to cleave genomic DNA and narrow down specific regions of the cleaved genome fragments for DNA sequencing [49,50,51]. GBS is a suitable method for genetic mapping, as it reduces the number of over-distributed SNPs in the genome and produces a large amount of genotypic information in a short period and at a low cost [49,50,51].

In this study, we aimed to identify QTLs conferring resistance to bacterial wilt caused by three different isolates of *R. solanacearum* in the F_2_ populations derived from self-pollination of the highly resistant pepper cultivar ‘Konesian Hot’ using GBS analysis.

## 2. Results

### 2.1. Bioassay of Bacterial Wilt Resistance Using Three Different Isolates of Ralstonia solanacearum

A bioassay was performed using three isolates of *R. solanacearum*, ‘IS’, ‘HS’, and ‘HWA’, with different pathogenicity in the F_2_ populations from a highly resistant pepper cultivar, ‘Konesian Hot’. Out of 92 individuals of the F_2_ population inoculated with the weakly pathogenic isolate ‘IS’, 91 individuals were resistant (disease index, DI, = 0 or 1) and only one individual was susceptible (DI = 2) (Table 1 and Figure 1 and Appendix A). The expected segregation ratio might be 1:0 since the observed ratio was not fitted to the expected ratio of 15:1 with the chi-square test at the level of 95% (Table 1). For the inoculation with the moderately pathogenic isolate ‘HS’, 54 individuals were resistant (DI = 0 or 1) and 42 were susceptible (DI = 2 or 3), and the ratio was fitted to the expected ratio of 9:7 with the chi-square test at the level of 95% (Table 1 and Figure 1 and Appendix A). After inoculation with the highly pathogenic isolate ‘HWA’, 55 individuals were found to be resistant (DI = 0 or 1) and 41 were susceptible (DI = 2 or 3), and the ratio was fitted to the expected ratio of 9:7 with the chi-square test at the level of 95% (Figure 1 and Table 1).

### 2.2. GBS Analysis of Two F_2_ Populations Derived from the Cultivar ‘Konesian Hot’ with Strong Resistance to Bacterial Wilt

A total of 96 individuals of the F_2_ population inoculated with the moderately pathogenic isolate ‘HS’ were subjected to GBS analysis for SNP discovery and QTL identification. A sequencing system Illumina HiSeq 2500 paired-end read was used for NGS. Sequencing of two lanes generated a total of 59.8 Gbp of raw sequences and approximately 592 million raw reads (Table 2). The raw sequences (KBRS20191025_0000001 and KBRS20191025_0000002) were deposited in the Korean BioInformation Center (https://www.kobic.re.kr/kobic/; accessed on 8 April 2022). The raw data were classified into 96 samples through a demultiplexing process using barcode sequences. Approximately 579 million raw reads (97.7%) were demultiplexed and, on average, there were 6,040,477 demultiplexed reads per sample (Table 2). Barcodes and adaptor sequences were removed in the demultiplexed sequences, and then the sequences were trimmed by the quality. The total length of trimmed reads was about 40.0 Gbp, which was 67.0% of the total length of raw reads (Table 2). The average length of trimmed reads per sample was 82.7 bp, and that of trimmed/raw data was 81.9% on average. The clean reads were mapped to the pepper reference genome (*C. annuum* cv. CM334 ver. 1.55; http://peppergenome.snu.ac.kr/; accessed on 8 April 2022). The total number of mapped reads was 429,461,067, which was 72.5% of the total number of raw reads (Table 2). The average number of mapped regions was 108,919, and the average depth and length of each mapped region were 17.27 and 144.32 bp, respectively. This covers 0.5739% of the reference genome on average (Table 2).

A total of 96 individuals of the F_2_ population inoculated with the highly pathogenic isolate ‘HWA’ were used for SNP identification and genotyping by GBS analysis, as described in the above paragraph. The Illumina HiSeq X paired-end read method was used for GBS sequencing (Table 2). Sequencing of two lanes generated approximately 109.5 Gbp of sequences (Table 2). The raw sequences were classified into 96 samples through a demultiplexing process using barcode sequences. About 630 million raw reads (87.0%) were demultiplexed, and the average number of demultiplexed reads per sample was 6,571,625 (Table 2). Through the trimming process, 65.3 Gbp of trimmed sequences was obtained, which was 59.7% of the total length of raw reads (Table 2). The average length of trimmed reads per sample was 113.78 bp, and the proportion of trimmed/raw reads was 75.4% on average. A total of 491,518,788 reads, 67.7% of the total number of raw reads, were mapped to the reference genome (Table 2). The average number of the mapped region was 112,352, and the average depth and length of each mapped region were 14.80 and 266.14 bp, respectively, while the reference genome coverage was 1.1020% (Table 2).

### 2.3. Selection of SNPs Identified by GBS Analysis in Two F_2_ Populations

In the ‘HS’-inoculated population, an SNP matrix was generated for the 96 samples. In total, 218,626 SNPs were detected with a criterion of minimum depth of 3, and 63,421 SNPs remained with the minor allele frequency (MAF) of over 5%. Finally, 12,227 SNPs were selected with less than 30% missing data.

In the ‘HWA’-inoculated population, an SNP matrix was made for 96 F_2_ individuals. A total of 434,974 SNPs were identified with a filtering criterion of minimum depth of 3. Only 19,044 SNPs were selected when the MAF and missing data were set to >5% and <30%, respectively.

### 2.4. Genetic Linkage Mapping of Pepper in Two F_2_ Populations

A genetic linkage map of an F_2_ population inoculated with the moderately pathogenic isolate ‘HS’ was constructed using the SNP matrix by GBS analysis and JoinMap^®^ ver. 4 (Appendix A). The SNPs unfitted to the 1:2:1 segregation ratio by χ^2^-test were excluded. A pepper genetic linkage map, composed of 17 linkage groups and 1120 SNP markers including 51 high-resolution melting (HRM) markers, was constructed using the SNP data (Table 3 and Appendix A). The total genetic distance and the average marker interval were 2432.8 cM and 2.17 cM/marker, respectively (Table 3). The number of markers per chromosome ranged from 65 to 129, with an average of 93.3 markers per chromosome (Table 3). The shortest linkage distance was 154.8 cM, for chromosome 11, and the longest was 267.0 cM, for chromosome 3 (Table 3).

A genetic linkage map for an F_2_ population inoculated with the highly pathogenic isolate ‘HWA’ was constructed with the same method described above (Appendix A). A pepper genetic linkage map, composed of 12 linkage groups and 1235 SNP markers including 31 HRM markers, was constructed (Table 3 and Appendix A). The total genetic distance and the average distance per linkage group were 2314.6 cM and 192.9 cM, respectively (Table 3). The number of markers per linkage group ranged from 70 to 148, with an average marker interval of 1.87 cM (Table 3). The shortest linkage group was chromosome 8 (160.1 cM), and the longest was chromosome 3 (246.0 cM) (Table 3).

### 2.5. Identification of QTLs for Bacterial Wilt Resistance in Two F_2_ Populations

QTL analysis was carried out based on the genotypic data by GBS analysis and the phenotypic data by DI scores using the composite interval mapping (CIM) method of Windows QTL Cartographer ver. 2.5. In the ‘HS’-inoculated population, seven candidate QTLs were detected on pepper chromosomes 2, 3, 5, 7, 8, 9, and 10. We developed 134 HRM markers in the vicinity of the detected candidate QTLs, of which 51 HRM markers were mapped on the pepper genetic linkage map (Table 3 and Appendix A).

QTL analysis revealed five QTLs, *Bwr6w-7.2*, *Bwr6w-8.1*, *Bwr6w-9.1*, *Bwr6w-9.2*, and *Bwr6w-10.1*, conferring the resistance to the moderately pathogenic ‘HS’ isolate (Figure 2 and Table 4). Additionally, four HRM markers, C07_224926788-HRM, C08_134064617-HRM, C09_3486004-HRM, and C10_232244800-HRM, were identified to be closely linked to the detected QTLs *Bwr6w-7.2*, *Bwr6w-8.1*, *Bwr6w-9.2*, and *Bwr6w-10.1*, respectively (Figure 2 and Table 4).

The QTL *Bwr6w-7.2*, closely linked to the C07_224926788-HRM marker, was detected with a logarithm of the odds (LOD) score of 3.28 and *R^2^* value of 13.05% (Table 4). *Bwr6w-8.1* was positioned on 105.7 cM of chromosome 8 with an LOD score of 3.28 and phenotypic variance of 12.67% (Figure 2 and Table 4). Two QTLs, *Bwr6w-9.1* and *Bwr6w-9.2*, were located between the C09_3440692 and C09_1831460 markers and between the C09_2974163-HRM and C09_6048971 markers with LOD scores of 3.53 and 3.37 and *R^2^* values of 15.07% and 10.46%, respectively (Figure 2 and Table 4). The QTL *Bwr6w-10.1* was closely linked to the C10_232244800-HRM marker, with an LOD score of 3.12 and phenotypic variance of 9.69% (Table 4).

In the ‘HWA’-inoculated population, QTL analysis showed three candidate QTLs on pepper chromosomes 5, 7, and 9. A total of 79 HRM markers were developed in the vicinity of the detected candidate QTLs, of which 31 markers were mapped on the pepper genetic linkage map (Table 3 and Appendix A). QTL analysis revealed three QTLs, *Bwr6w-5.1*, *Bwr6w-6.1*, and *Bwr6w-7.1*, conferring the resistance to the highly pathogenic isolate ‘HWA’ (Figure 3 and Table 4). Additionally, two HRM markers, C05_224016474-HRM and C07_115436147-HRM, were identified to be closely linked to the detected QTLs *Bwr6w-5.1* and *Bwr6w-7.1*, respectively (Figure 3 and Table 4).

The QTL *Bwr6w-5.1* was located between the markers C05_224044238 and C05_224047094 on chromosome 5 with an LOD score of 5.89 and *R^2^* value of 19.67% (Table 4). The QTL *Bwr6.1* was positioned between the markers C06_200031568 and C06_203415605 on chromosome 6 showing an LOD score of 3.78 and phenotypic variance of 16.50% (Table 4). The QTL *Bwr6w-7.1* was closely linked to the C07_115436147-HRM marker on chromosome 7 with an LOD score of 3.27 and *R^2^* value of 12.56% (Figure 3 and Table 4).

Eight QTL-linked markers including six HRM markers (Appendix A) and two GBS-based SNP markers were further subjected to analyze the genotypic effect of markers on bacterial wilt resistance (Figure 4). The result showed that, in all markers, B genotypes, derived from the paternal parent, were more highly resistant to bacterial wilt than A genotypes (Figure 4). H and B genotypes of a marker, C07_224926788-HRM, linked to the QTL *Bwr6w-7.2* showed more resistance than the A genotype (Figure 4A). H and B genotypes of a marker, C08_134064617-HRM, linked to the QTL *Bwr6w-8.1* showed more resistance than the A genotype (Figure 4B). An SNP, C09_2427860, linked to *Bwr6w-9.1* showed an additive effect on mean disease index depending on genotypes (Figure 4C). A marker, C09_3486004-HRM, linked to *Bwr6w-9.2* showed an additive effect on mean disease index depending on genotypes (Figure 4D). A marker, C10_232244800-HRM, linked to *Bwr6w-10.1* showed an additive effect on mean disease index depending on genotypes (Figure 4E). A marker, C05_224016474-HRM, linked to *Bwr6w-5.1* showed an additive effect on mean disease index depending on genotypes (Figure 4F). The B genotype of the SNP C06_204307935 linked to the QTL *Bwr6w-6.1* showed more resistance than A and H genotypes (Figure 4G). The B genotype of the marker C07_115436147-HRM linked to the QTL *Bwr6w-7.1* showed more resistance than the A and H genotypes (Figure 4H).

## 3. Discussion

Bacterial wilt, caused by the soil-borne pathogen *R. solanacearum*, is one of the most serious diseases of chili pepper (*C. annuum* L.) worldwide, predominantly occurring in tropical, subtropical, and temperate regions [9,10,11]. The economic loss due to bacterial wilt is becoming more serious in South Korea, where average summer temperatures rise every year [4]. To date, there is no effective chemical control method for *R. solanacearum*; hence, development of varieties with sustainable resistance to bacterial wilt is needed [39,41]. Despite the many achievements in resistance breeding over the last three decades, commercial pepper varieties resistant to bacterial wilt are not widespread in Korea [47]. The use of molecular markers will substantially reduce the cost and time required for the development and release of pepper varieties resistant to bacterial wilt disease [42,43,44].

The hybrid variety ‘Konesian Hot’ used for stock was resistant to 14 isolates of *R. solanacearum* in South Korea [4]. The variety showed strong resistance to all the strains (groups II to VI) except for the highly pathogenic group (group I) [4]. *R**alstonia solanacearum* is heterogeneous among strains and highly diversified among groups of strains [25,26,27]. The resistance to *R. solanacearum* in pepper is known to be a quantitative trait [45,46,47]. However, there are no studies that have evaluated the differences of pepper resistance loci depending on isolates of *R. solanacearum*. In this study, three F_2_ populations derived from self-pollination of the highly resistant pepper cultivar ‘Konesian Hot’ were evaluated for resistance to three isolates of *R. solanacearum* with different pathogenicity, ‘IS’, ‘HS’, and ‘HWA’, (Figure 1 and Table 1). In the ‘HS’- and ‘HWA’-inoculated populations, two pepper genetic linkage maps were constructed through GBS analysis (Appendix A), and QTLs for bacterial wilt resistance were identified using a CIM method (Figure 2 and Figure 3 and Table 4). In addition, HRM markers closely linked to the QTLs were developed (Figure 2 and Figure 3 and Table 4).

To date, three QTL analyses for bacterial wilt resistance have been reported [44,46,47]. A QTL, *Bw1*, linked to an SSR marker, CAMS451, was detected on pepper chromosome 1 with phenotypic variance of 33% [44]. A major QTL, *qRRs-10.1*, was identified to account for 19.01% of the phenotypic variation and to be located between the ID10-194305124 and ID10-196208712 markers on pepper chromosome 10 [46]. Recently, a QTL, *pBWR-1*, was found on chromosome 1, explaining 20.13 to 25.16% of *R^2^* values [47]. However, since the developed molecular markers are not widely used to develop new cultivars with bacterial wilt resistance, we aimed to develop practical SNP markers closely linked to the QTLs.

In the bioassay of an F_2_ population inoculated with a weakly pathogenic isolate, ‘IS’, of *R. solanacearum*, no segregation of phenotypes was observed except for a single plant that had a disease index of 2, implying that both parents have the gene resistant to the ‘IS’ isolate (Figure 1 and Table 1). In contrast, in the inoculation with the moderately pathogenic isolate ‘HS’, the phenotypic segregation ratio was 9:7 (54 resistant and 42 susceptible), suggesting the possible involvement of two complementary genes (Table 1). In the F_2_ population inoculated with the highly pathogenic isolate ‘HWA’, the phenotypic segregation ratio was also 9:7 (55 resistant and 41 susceptible), indicating that two dominant resistance genes were involved (Table 1). These results were consistent with previous studies [4,8], showing that even within a given population, the segregation of resistance to bacterial wilt varies depending on isolates of *R. solanacearum* (Figure 1 and Table 1).

GBS is a method suitable for SNP detection and genotyping, with fast construction of genetic linkage maps [48,49,50,51]. Using the genotypic data from GBS analysis and the phenotypic data by disease index scores, we identified eight QTLs conferring bacterial wilt resistance in two segregating F_2_ populations of *C. annuum*, ‘Konesian Hot’ (Table 4). Five QTLs, *Bwr6w-7.2*, *Bwr6w-8.1*, *Bwr6w-9.1*, *Bwr6w-9.2*, and *Bwr6w-10.1*, were identified on pepper chromosomes 7, 8, 9, and 10 for resistance to the moderately pathogenic isolate ‘HS’, and three QTLs, *Bwr6w-5.1*, *Bwr6w-6.1*, and *Bwr6w-7.1*, were detected on chromosomes 5, 6, and 7 for resistance to the highly pathogenic isolate ‘HWA’ (Table 4). Additionally, six HRM markers closely linked to the QTLs were developed: C07_224926788-HRM, C08_134064617-HRM, C09_3486004-HRM, C10_232244800-HRM, C05_224016474-HRM, and C07_115436147-HRM, (Figure 2, Figure 3, and Appendix A).

The QTLs *Bwr6w-10.1* and *qRRs-10.1* were detected on the same chromosome, implying the same QTL [46]. Therefore, further studies are required. In this study, the QTL on chromosome 1 was not found to be inconsistent with the findings of previous studies that the QTLs *Bw1* and *pBWR-1* were located on chromosome 1 [44,47]. These results suggest that the resistance to *R. solanacearum* in pepper might differ depending on the isolates of *R. solanacearum* or plant materials, etc. [45], because the resistance is controlled by polygenic genes in a quantitative form [5,45]. To date, this study found the highest number of QTLs conferring bacterial wilt resistance in pepper.

Unveiling QTL information and developing closely linked markers could accelerate the development of highly resistant pepper varieties to bacterial wilt [42,43,44,46,47]. Additionally, this QTL information could narrow down candidate resistance genes and result in molecular cloning and functional characterization of the resistance genes using transformation and genome editing technologies [52].

## 4. Materials and Methods

### 4.1. Plant Materials

According to a report by Lee et al. [4], four resistance levels were reported with varying degrees of resistance to *R. solanacearum* isolates collected from major cultivated areas of chili pepper in Korea, out of which ‘Konesian Hot’ was strongly resistant to bacterial wilt. In order to understand the genetic basis of bacterial wilt resistance, three sets of 96 segregating F_2_ populations derived from the self-pollination of a hybrid (F_1_) cultivar ‘Konesian Hot’ were used (Figure 1 and Table 1). For the bioassay, a pepper cultivar ‘Konesian Hot’ was used as a resistant control, and a pepper cultivar ‘Geonchowang’, which was susceptible to bacterial wilt, was used as a susceptible control [4]. All the F_2_ plants used in the experiment were used for DNA extraction for GBS analysis.

### 4.2. Pathogen Materials and Inoculum Preparation

A bioassay was performed using three isolates of *R. solanacearum* with different pathogenicity, ‘IS’, ‘HS’, and ‘HWA’ (Figure 1 and Table 1). The isolates used in the experiments were from the work of Lee et al. [4]. The three different isolates of *R. solanacearum* included a weakly pathogenic isolate ‘IS’ (group VI, race 1, and biovar 4), a moderately pathogenic isolate ‘HS’ (group III, race 1, and biovar 4), and a highly pathogenic isolate ‘HWA’ (group I, race 1, and biovar 4) [4].

The pathogens were cultured by the following method. The NA medium (nutrient agar, ordinary agar 20 g; distilled water 1 L) streaked with an isolate was incubated in a chamber at 28 °C for 48 h. The bacterial ooze cultured on an NA medium was dissolved 20–30 times in LB liquid medium (Luria–Bertani broth 25 g; distilled water 1L) using a loop and then cultured for 48 h on a shaking incubator (28 °C, 147 rpm). Absorbance of cultured pathogens was measured at a wavelength of 600 nm using a spectrophotometer. The concentration was adjusted to 10^8^ colony-forming units (CFU)·mL^−1^ (OD_600nm_ = 0.4~0.5) and used as inoculum [8].

### 4.3. Evaluation of Bacterial Wilt Resistance

The evaluation of bacterial wilt resistance was performed according to Tran and Kim [8] and Lee et al. [4]. At 30 days after sowing the pepper seeds (4–6 sheets of foliage leaf), the lateral root was pruned with sterilized pincettes, and then each plant was inoculated with 5 mL of the prepared inoculum. As bacterial wilt is well developed under high temperature and humidity conditions, the temperature in the greenhouse was managed so that it would not drop below 25 °C after inoculation. Symptoms caused by *R. solanacearum* in pepper were classified into four DI scores, 0 for no symptoms and healthy plant, 1 for cotyledon yellowing and etiolated, 2 for most of the leaves and stem wilted, and 3 for the whole plant wilted or dead (Appendix A). At the sixth week after inoculation, the resistance was classified as resistant for DI = 0 or 1 and as susceptible for DI = 2 or 3 (Table 1 and Figure 1).

### 4.4. DNA Extraction

Genomic DNA was extracted from young leaves of pepper F_2_ individuals according to the method described by Lee et al. [53]. Young leaves of pepper were placed in a 2.0 mL microcentrifuge tube containing three 4 mm stainless steel beads and 800 μL of DNA-extraction buffer (20 mM Tris-HCI, pH = 7.5; 250 mM NaCl; 25 mM EDTA; 0.5% SDS), 1 g polyvinylpyrrolidone (PVP), and 14 μL β-mercaptoethanol, shaken at 30 Hz for 3 min using a Tissue Lyser Ⅱ (Qiagen, Hilden, Germany). After heat treatment for 10 min at 70 ℃ in a Lab Armor bead bath (Thermo Fisher Scientific Inc., Waltham, MA, USA), centrifugation was performed at 13,000 rpm for 10 min at 4 °C using a 1730R centrifuge (Labogene, Seoul, Korea). Then, 700 μL of supernatant was transferred to a new 2.0 mL microcentrifuge tube and mixed with 700 μL of a ratio of chloroform to isoamyl alcohol (24:1). The mixture was centrifuged for 10 min at 13,000 rpm at 4 °C. Next, 600 μL of supernatant was transferred into a new 1.5 mL microcentrifuge tube and mixed with 600 μL of isopropanol. This mixture was held for more than 30 min at −20 °C and then centrifuged for 10 min at 13,000 rpm at 4 °C. The supernatant was removed, and the pellet was washed twice using 500 μL of 70% ethanol and centrifuged for 1 min at 13,000 rpm at 4 °C. Finally, after drying the pellet well, the DNA was dissolved in 100 μL of distilled water and treated with 0.1 μL (10 mg·mL^−1^) of RNase solution (Bio Basic Canada Inc., Markham, ON, Canada). The DNA concentration was measured using a BioDrop LITE (BioDrop UK Ltd., Cambridge, UK).

### 4.5. GBS Analysis

Genomic DNA from two sets of 96 F_2_ individuals was used to construct libraries for GBS analysis, conducted by the bioinformatics company SEEDERS Co. (Daejeon, South Korea). The first set of F_2_ population was inoculated with the moderately pathogenic isolate ‘HS’, and the second set of F_2_ population was inoculated with the highly pathogenic isolate ‘HWA’ (Table 2). GBS library was constructed using restriction enzymes, *Ape*KI. The first set library was then pooled and sequenced with a paired-end read method using HiSeq 2500 (Illumina, San Diego, CA, USA), and the second set was sequenced using HiSeq X (Illumina) for GBS sequencing (Table 2). Demultiplexing was performed by 96 barcode sequences. Adapter trimming was performed using the Cutadapt ver. 1.8.3 program [54], and the DynamicTrim and LengthSort programs of the SolexaQA ver. 1.13 package [55] were used for sequence quality trimming. Clean reads that passed the pretreatment process were mapped to the reference genome (*C. annuum* cv. CM334 ver. 1.55) available on the Sol Genomics Network (http://www.sgn.cornell.edu/; accessed on 8 April 2022) using the Burrows–Wheeler Alignment (BWA) ver. 0.6.1-r104 program [56]. The clean reads mapped to the reference genome were searched for raw SNPs through the SAMtools ver. 0.1.16 program [57], and then the consensus sequence was extracted. SNPs were validated using a SEEDERS in-house script [58] and used to generate an integrated SNP matrix.

### 4.6. Genetic Linkage Map Construction

Genetic linkage maps were constructed using JoinMap ver. 4.1 (Kyazma B.V., Wageningen, The Netherlands). A logarithm of odds (LOD) score of 3.0 and a maximum distance of 30 cM were used as thresholds to determine the significant linkage between markers. Genetic map distance (cM) was calculated using the Kosambi mapping function [59]. The final linkage map was drawn using the MapChart ver. 2.2 program [60].

### 4.7. QTL Analysis

The QTL analysis was conducted using Windows QTL Cartographer ver. 2.5 [61,62,63,64]. To recognize the association between SNP markers and bacterial wilt resistance, composite interval mapping (CIM) was used with 2.0 cM walk speed. The genome-wide LOD threshold for significance level (*p* = 0.05) was estimated by 1000 permutation tests.

### 4.8. Primer Design for HRM Analysis

Primer sets were designed using the Primer3 ver. 0.4.0 software program (https://bioinfo.ut.ee/primer3-0.4.0/; accessed on 8 April 2022) (Appendix A). To analyze the presence of homologs, the amplicons were blasted to the pepper reference genome (*C. annuum* cv. CM334 ver. 1.55) published on the Pepper Genome Platform (http://peppergenome.snu.ac.kr/; accessed on 8 April 2022).

### 4.9. PCR and HRM Analysis

PCR reaction for HRM analysis was prepared using 20 ng·μL^−1^ genomic DNA, 10× PCR buffer, 2.5 mM dNTP mixture, 0.1 units *Taq* DNA polymerase (TransGen Biotech Co., Beijing, China), 0.5 μL SYTO^®^ 9 green, fluorescent nucleic acid stain (Life Technologies^TM^, Carlsbad, CA, USA), and 10^−5^ μL of each primer in a final reaction volume adjusted to 20.0 μL, using TDW. Biometra TAdvanced (Analytik Jena AG, Jena, Germany) was used for the PCR reaction. After performing the initial denaturation at 95 °C for 5 min, denaturation at 95 °C for 10 s and the annealing/elongation process at 60 °C for 20 s were repeated 39 times. Next, a full extension reaction was carried out at 72 °C for 20 s. The PCR product was used to create a melting curve using LightCycler^®^ Real Time PCR (Roche Holding AG, Basel, Switzerland). The fluorescence value of SYTO^®^ 9 was measured at each temperature while raising the temperature by 0.03% from 65 °C to 97 °C. The melting curve graph was analyzed using High-Resolution Melt software v.1.1 (Roche Holding AG, Basel, Switzerland) for genotype analysis. Genotypes were classified into three groups: A (maternal homozygous), B (paternal homozygous), and H (heterozygous) (Appendix A).

### 4.10. Statistical Analysis

Phenotypic segregation ratio was analyzed by the chi-square (χ^2^) test. Statistical significance was identified by Duncan’s multiple range test at 5% level using R package ver. 3.6.3.

## 5. Conclusions

In this study, we identified eight significant QTLs resistant to bacterial wilt in the F_2_ population obtained from self-pollination of the highly resistant pepper cultivar ‘Konesian Hot’ using GBS analysis. QTL analysis revealed five QTLs, *Bwr6w-7.2*, *Bwr6w-8.1*, *Bwr6w-9.1*, *Bwr6w-9.2*, and *Bwr6w-10.1*, conferring resistance to the moderately pathogenic isolate ‘HS’ and three QTLs, *Bwr6w-5.1*, *Bwr6w-6.1*, and *Bwr6w-7.1*, showing resistance to the highly pathogenic isolate ‘HWA’. Additionally, six HRM markers closely linked to the QTLs, C07_224926788-HRM, C08_134064617-HRM, C09_3486004-HRM, C10_232244800-HRM, C05_224016474-HRM, and C07_115436147-HRM, were developed. The QTL information and HRM markers will accelerate the development of pepper varieties with resistance to bacterial wilt.

## Figures and Tables

**Figure 1 plants-11-01551-f001:**
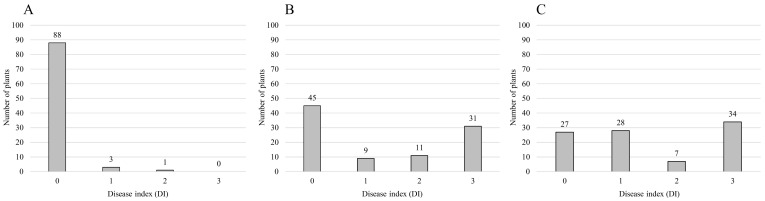
Frequency distribution of disease index (DI) caused by (**A**) a weakly pathogenic isolate ‘IS’, (**B**) a moderately pathogenic isolate ‘HS’, and (**C**) a highly pathogenic isolate ‘HWA’ of *Ralstonia solanacearum* in three F_2_ populations of a resistant pepper cultivar ‘Konesian Hot’.

**Figure 2 plants-11-01551-f002:**
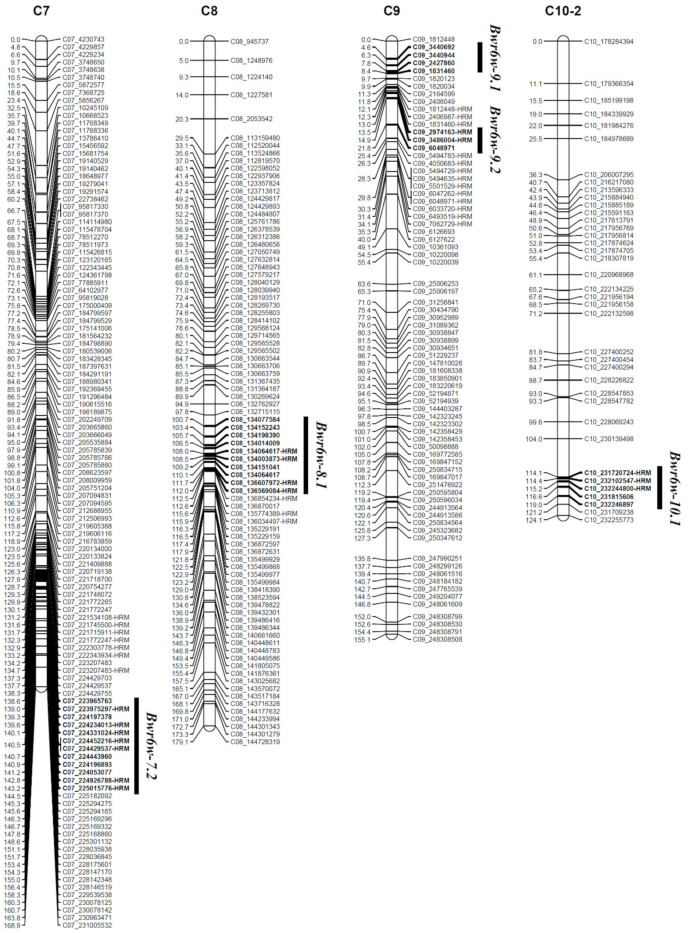
Map position of QTLs for the resistance to bacterial wilt caused by a moderately pathogenic ‘HS’ isolate of *Ralstonia solanacearum* in an F_2_ population of the strongly resistant hybrid pepper cultivar ‘Konesian Hot’. Bar left number, genetic position (cM); bar right name, name of SNP marker; bold, markers located within each QTL region.

**Figure 3 plants-11-01551-f003:**
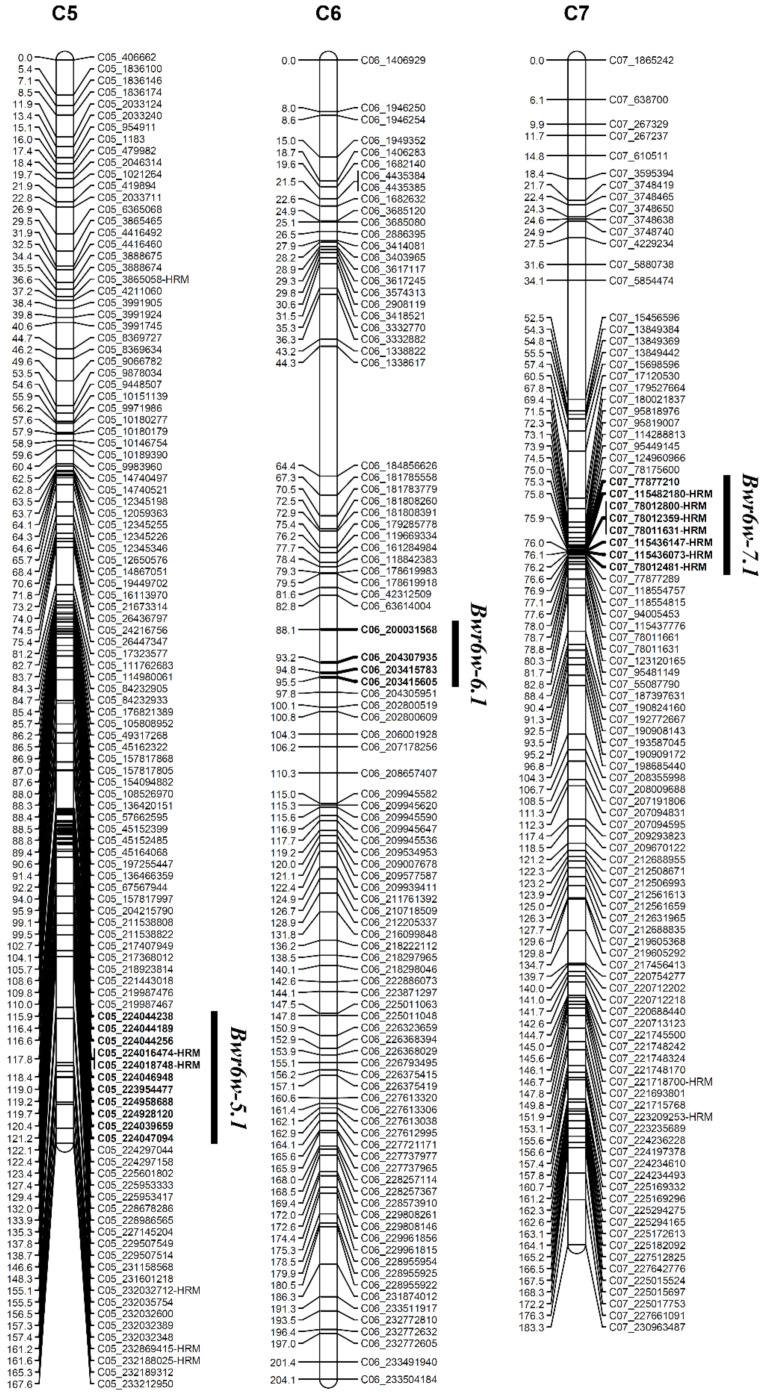
Map position of QTLs for the resistance to bacterial wilt caused by a highly pathogenic ‘HWA’ isolate of *Ralstonia solanacearum* in an F_2_ population of the strongly resistant hybrid pepper cultivar ‘Konesian Hot’. Bar left number, genetic position (cM); bar right name, name of SNP marker; bold, markers located within each QTL region.

**Figure 4 plants-11-01551-f004:**
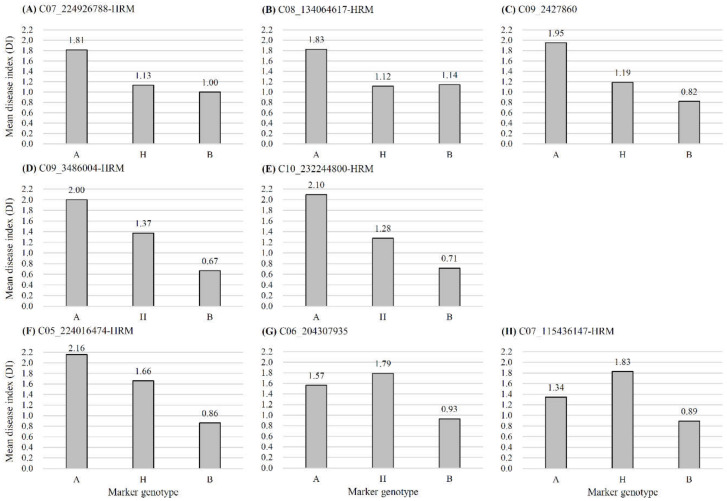
Comparison of mean disease index (DI) for bacterial wilt caused by ‘HS’ (**A**–**E**) and ‘HWA’ (**F**–**H**) isolates of *Ralstonia solanacearum* between genotypes of the HRM or SNP markers closely linked to one of the eight QTLs detected in this study. A, maternal genotype; B, paternal genotype; H, heterozygous genotype. (**A**) H and B genotypes of a marker, C07_224926788-HRM, linked to the QTL *Bwr6w-7.2* showed more resistance than A genotype. (**B**) H and B genotypes of a marker, C08_134064617-HRM, linked to the QTL *Bwr6w-8.1* showed more resistance than A genotype. (**C**) An SNP, C09_2427860, linked to *Bwr6w-9.1* showed an additive effect on mean disease index depending on genotypes. (**D**) A marker, C09_3486004-HRM, linked to *Bwr6w-9.2* showed an additive effect on mean disease index depending on genotypes. (**E**) A marker, C10_232244800-HRM, linked to *Bwr6w-10.1* showed an additive effect on mean disease index depending on genotypes. (**F**) A marker, C05_224016474-HRM, linked to *Bwr6w-5.1* showed an additive effect on mean disease index depending on genotypes. (**G**) B genotype of SNP C06_204307935 linked to the QTL *Bwr6w-6.1* showed more resistance than A and H genotypes. (**H**) B genotype of a marker, C07_115436147-HRM, linked to the QTL *Bwr6w-7.1* showed more resistance than A and H genotypes.

**Table 1 plants-11-01551-t001:** Segregation analysis of bacterial wilt resistance in three F_2_ populations of the highly resistant pepper cultivar ‘Konesian Hot’ inoculated with three different isolates of *Ralstonia solanacearum*.

Population	Isolate of *Ralstonia solanacearum*^z^	Number of Plants			Segregating Ratio	Chi-Square Value	Probability Value
		Resistant (DI^y^ = 0 or 1)	Susceptible (DI = 2 or 3)	Total			
F_2_ of ‘Konesian Hot’	‘IS’	91	1	92	15:1	4.186	0.041
F_2_ of ‘Konesian Hot’	‘HS’	54	42	96	9:7	0	1
F_2_ of ‘Konesian Hot’	‘HWA’	55	41	96	9:7	0.042	0.837

**Table 2 plants-11-01551-t002:** Summary of sequence data generated by genotyping-by-sequencing analysis using two F_2_ populations of ‘Konesian Hot’ inoculated with a moderately pathogenic ‘HS’ isolate and a highly pathogenic ‘HWA’ isolate of *Ralstonia solanacearum*.

GBS Statistics	‘HS’-Inoculated Population	‘HWA’-Inoculated Population
Number of F_2_ plants for multiplexing	96	96
Sequencing system	Illumina HiSeq 2500	Illumina HiSeq X
Average length of raw reads (bp)	101 (100%)	151 (100%)
Total number of sequenced raw reads	592,251,242 (100%)	725,489,528 (100%)
Total length of sequenced raw data (bp)	59,817,375,442 (100%)	109,548,918,728 (100%)
Total number of demultiplexed reads	579,885,814 (97.7%)	630,875,996 (87.0%)
Average number of demultiplexed reads/sample	6,040,477	6,571,625
Total length of demultiplexed reads (bp)	58,568,467,214 (97.9%)	95,262,275,396 (87.0%)
Number of trimmed reads	484,089,570 (81.7%)	574,385,318 (79.2%)
Total length of trimmed reads (bp)	40,076,781,039 (67.0%)	65,375,963,492 (59.7%)
Average length of trimmed reads (bp)	82.7 (81.9%)	113.78 (75.4%)
Total length of pepper reference genome (*Capsicum annuum* cv. CM334 ver.1.55) (bp)	2,753,501,687	2,753,501,687
Number of mapped reads	429,461,067 (72.5%)	491,518,788 (67.7%)
Average number of mapped regions	108,919	112,352
Average depth of mapped region	17.27	14.80
Average length of mapped region (bp)	144.32	266.14
Reference genome coverage (%)	0.5739	1.1020

**Table 3 plants-11-01551-t003:** Summary of two pepper genetic linkage maps generated by GBS analysis.

Chr.^z^ No.	‘HS’-Inoculated Population					‘HWA’-Inoculated Population				
	Number of Markers			Length of Linkage Distance (cM)	Average Marker Interval (cM/marker)	Number of Markers			Length of Linkage Distance (cM)	Average Marker Interval (cM/marker)
	GBS^y^- Based SNP	HRM^x^	Total			GBS-Based SNP	HRM	Total		
1	95	0	95	308.8	3.25	107	0	107	184.1	1.72
2	90	2	92	184.9	2.01	100	1	101	172.8	1.71
3	125	4	129	267.0	2.07	145	2	147	246.0	1.67
4	87	0	87	185.7	2.13	70	0	70	195.1	2.79
5	61	4	65	185.6	2.86	108	6	114	167.6	1.47
6	91	0	91	207.0	2.27	96	0	96	204.1	2.13
7	103	14	117	168.9	1.44	92	9	101	183.3	1.81
8	78	7	85	179.1	2.11	81	3	84	160.1	1.91
9	60	15	75	155.1	2.07	80	4	84	193.2	2.30
10	65	3	68	229.5	3.38	99	2	101	203.4	2.01
11	124	0	124	154.8	1.25	148	0	148	193.2	1.31
12	90	2	92	206.5	2.24	78	4	82	211.7	2.58
Total	1069	51	1120	2432.8	2.17	1204	31	1235	2314.6	1.87

**Table 4 plants-11-01551-t004:** Summary of QTLs for pepper resistance to bacterial wilt caused by two different isolates, a moderately pathogenic ‘HS’ and a highly pathogenic ‘HWA’, of *Ralstonia solanacearum*.

Trait	QTL	Chr.	QTL Peak Position (cM)	Marker Interval	The Closest HRM Marker	*R^2^* (%)	Additive	Dominant	LOD	LOD Threshold
‘HS’ resistance	*Bwr6w-7.2*	7	141.2	C07_223965763-C07_225015776-HRM	C07_224926788-HRM	13.05	0.4251	−0.5439	3.28	3.1
	*Bwr6w-8.1*	8	105.7	C08_134077584-C08_136569084-HRM	C08_134064617-HRM	12.67	0.3439	−0.6937	3.28	3.1
	*Bwr6w-9.1*	9	7.3	C09_3440692-C09_1831460	-	15.07	0.6336	−0.2142	3.53	3.1
	*Bwr6w-9.2*	9	14.9	C09_2974163-HRM-C09_6048971	C09_3486004-HRM	10.46	0.6555	−0.0027	3.37	3.1
	*Bwr6w-10.1*	10	117.7	C10_231720724-HRM-C10_232246897	C10_232244800-HRM	9.69	0.6061	−0.0337	3.12	3.1
‘HWA’ resistance	*Bwr6w-5.1*	5	120.4	C05_224044238-C05_224047094	C05_224016474-HRM	19.67	0.7266	0.0500	5.89	3.0
	*Bwr6w-6.1*	6	93.2	C06_200031568-C06_203415605	-	16.50	0.4528	0.4926	3.78	3.0
	*Bwr6w-7.1*	7	76.0	C07_77877210-C07_78012481-HRM	C07_115436147-HRM	12.56	0.3109	0.7013	3.27	3.0

## Data Availability

Not applicable.

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
