# Peer review of "QTL Mapping for Resistance to Bacterial Wilt Caused by Two Isolates of Ralstonia solanacearum in Chili Pepper (Capsicum annuum L.)"

_plants, 2022, doi:10.3390/plants11121551_

Round 1

Reviewer 1 Report

This article attempts to understand the genetic basis of bacterial wilt disease caused by Ralstonia solanacearum in the chilli pepper by QTL identification. 

I have a following comments regarding the manuscript:

1. 92 F2 population for scoring all the three isolates pathogenicity is relatively lesser to draw a firm conclusion, as it is a segregating population and not  a RIL.

2. Why two different sequencing platforms were used for HWA and HS inoculated plants?

3.The genetic basis i.e. the genes involved in the regions mapped to the QTL (as described in table 4) are not described. Without this study, it is difficult to correlate the phenotype to the genotype and its susceptibility. This information would be useful in the future for the breeders to alter genes and study the functional aspect of the genes involved.

Author Response

1. 92 F2 population for scoring all the three isolates pathogenicity is relatively lesser to draw a firm conclusion, as it is a segregating population and not a RIL.

Authors’ reply> We used three F2 populations which consisted 96 individuals per population. RIL is an ideal population but it takes for a long time to construct RIL population. For a firm conclusion, we will use various segregating populations.

2. Why two different sequencing platforms were used for HWA and HS inoculated plants?

Authors’ reply> We wanted to know the sequencing efficiency according to sequencing platforms, HiSeq 2500 and HiSeq X. HiSeq X was more efficient than HiSeq 2500.

3.The genetic basis i.e. the genes involved in the regions mapped to the QTL (as described in table 4) are not described. Without this study, it is difficult to correlate the phenotype to the genotype and its susceptibility. This information would be useful in the future for the breeders to alter genes and study the functional aspect of the genes involved.

Authors’ reply> We agree with you. Candidate genes are very important information. Therefore, we are preparing for next manuscript for candidate genes involved in the regions mapped to the QTL.

Reviewer 2 Report

The manuscript entitled "QTL Mapping for Resistance to Bacterial Wilt Caused by Two Isolates of Ralstonia” talks about QTL mapping of resistance against bacterial wilt in pepper. Pepper bacterial wilt is a very destructive disease, and it is hard to control this disease by bactericidal pesticides. Therefore, it is meaningful to do germplasm exploration and genetic analysis, which do help to resistance breeding of pepper bacterial wilt. Importantly, this manuscript also develops and validates the linked molecular marker. There are already many literatures about QTL mapping of pepper bacterial wilt. The biggest novelty of this manuscript is the resistance gene resource “Konesian hot”. Basically, this is a report of typical primary QTL mapping but it seems very strange to use a F2 population without any information of parents. People may take this strategy to study those plants that are hard to make seeds with clearly known parents like tea trees and many fruit trees. This is not good and unnecessary to be applied on peppers. The authors probably do not have the parent plant materials. The signals of the detected QTLs are not strong, the detected QTLs are not supported by other evidences, and the language of this manuscript should be carefully refined and well-organized.

There are some suggestions:

(1) Table S1 and Table S2.

I found the writers provided a summary of the developed HRM makers. Where are the primer information? Academic sharing is an essential principle to achieve academic excellence. Moreover, a publication should provide enough information for others to repeat the experiments.

The raw data of GBS sequencing should also deposited in public database and provide access link or ID.

(2) page 3, line 62-64

‘Konesian Hot’ show resistance to five strains. Before this sentence, the authors introduced different methods to classify the pathogens, like “four phylogypes”, “eight clades”, and the most popular strain is viovar 4 of race 1. I hope the authors can state the resistance characteristic of ‘Konesian Hot’ clearly.

(3) page 3, line 71-3

It is unnecessary to introduce what is qualitative and quantitative traits. Please delete sentences like this.

(4) page 3, line 78-84

This paragraph is the most important one in the section of introduction. The authors can provide more information about the QTL mapping work of pepper wilting disease. The authors made three quotations here. In factor, there are many more literatures about this work. This introduction can be expanded.

(5)page 4, line 94-96

In the abstract, it is two isolates: moderate ‘HS’ isolate and more virulent isolate ‘HWA’.  Here, it is “three different isolates”. In section 2.1, three isolates.

(6) line 100

What is the meaning of IS, HS, HWA? What race or biovar should them belong to?

(7) figure1

The frequency of F1 resistance should be added. 100%?

The population is too small.

All the F2 seedlings show resistance to IS. IS is not virulent at all? There is no segregation of resistance in F2?

HS is a moderate virulent isolate, and WHS is a stronger isolate. From this result, we cannot see such a big difference. For both HS and HWA, the segregation is 9:7.

(8) line 108

“ (Supplementary Materials))”

(9) section 2.2

Why the generated reads size of ‘HWA’ is double of ‘HS’? Does this affect the QTL mapping results?

(10) line 170, line 173

“seven candidate QTLs”, “five QTLs”, there should be some explanation.

(11) line 207-208

“B genotype of SNP 208 C06_204307935 linked to the QTL Bwr6w-6.1 showed more resistance than A and H genotypes”. Why not check what gene is positioned nearby?

(12) section 2.5

It will be better if the QTLs detected by different pathogen isolate named differently.

HRM markers. Please show the pictures of melting curves for important markers stressed by the authors.

(13) line 222

“The hybrid variety ‘Konesian Hot’ used for stock”, this information is important. Please stress the significance why choose this cultivar and provide more information like this in M&M, even in section of introduction.

(14) line 234-line 240

What is the conclusion of this discussion. The QTLs identified in this study positioned at the same place with others? Different? How about comparing positions of QTLs between HS and HWA?

(15) line 367-368

Please show the picture of genotyping for important makers.

(16) table 2

Will the authors explain why use different sequencing platform?

(17)fig 2 and fig3

How about integrate QTL7.1 and QTL7.2 into one map?

(18)figure 4

Is there statistically significance? I think some markers are not valuable.

(19)section 4.3

What is the culture condition of seedlings used to do resistance assay? Pot? Cell tray? Field? The resistance investigation was performed at the 6th week after inoculation? Really. It is too long in my knowledge. We do this experiment at 10th day.

Author Response

(1) Table S1 and Table S2.

I found the writers provided a summary of the developed HRM makers. Where are the primer information? Academic sharing is an essential principle to achieve academic excellence. Moreover, a publication should provide enough information for others to repeat the experiments. The raw data of GBS sequencing should also deposited in public database and provide access link or ID.

Authors’ reply> We provided the primer information of HRM markers mapped in this study (Table S3 and Table S4). And we deposited the raw data in Korean BioInformation Center (page 4).

(2) page 3, line 62-64

‘Konesian Hot’ show resistance to five strains. Before this sentence, the authors introduced different methods to classify the pathogens, like “four phylogypes”, “eight clades”, and the most popular strain is viovar 4 of race 1. I hope the authors can state the resistance characteristic of ‘Konesian Hot’ clearly.

Authors’ reply> We added the sentence (page 2).

(3) page 3, line 71-3

It is unnecessary to introduce what is qualitative and quantitative traits. Please delete sentences like this.

Authors’ reply> We deleted the sentence (page 2).

(4) page 3, line 78-84

This paragraph is the most important one in the section of introduction. The authors can provide more information about the QTL mapping work of pepper wilting disease. The authors made three quotations here. In factor, there are many more literatures about this work. This introduction can be expanded.

Authors’ reply> We added the sentence for QTL mapping work (pages 2~3).

(5) page 4, line 94-96

In the abstract, it is two isolates: moderate ‘HS’ isolate and more virulent isolate ‘HWA’.  Here, it is “three different isolates”. In section 2.1, three isolates.

Authors’ reply> We used three isolates for resistance analysis and used two isolates for QTL analysis.

(6) line 100

What is the meaning of IS, HS, HWA? What race or biovar should them belong to?

Authors’ reply> They are isolate names, which belong to race 1, biovar 4.

(7) figure1

The frequency of F1 resistance should be added. 100%?

Authors’ reply> Fig.1 showed the resistance segregation in F2 populations. F1 plants all showed 0 of disease index.

The population is too small.

Authors’ reply> 96 plants are enough to construct a genetic linkage map.

All the F2 seedlings show resistance to IS. IS is not virulent at all? There is no segregation of resistance in F2?

Authors’ reply> IS isolate is not virulent at all to offspring of ‘Konesian Hot’.

HS is a moderate virulent isolate, and WHA is a stronger isolate. From this result, we cannot see such a big difference. For both HS and HWA, the segregation is 9:7.

Authors’ reply> There is not a big difference, but a little difference. The number of plants (45 plants) (disease index = 0) in ‘HS’ inoculated population was almost two-fold than the number (27 plants) in ‘HWA’ inoculated population.

(8) line 108

“ (Supplementary Materials))”

Authors’ reply> In this journal, it is permitted.

(9) section 2.2

Why the generated reads size of ‘HWA’ is double of ‘HS’? Does this affect the QTL mapping results?

Authors’ reply> It is caused by the sequencing machines, HiSeq 2500 vs. HiSeq X. QTL results are not affected by sequencing methods.

(10) line 170, line 173

“seven candidate QTLs”, “five QTLs”, there should be some explanation.

Authors’ reply> Seven candidate QTLs were detected on pepper genetic linkage map constructed by GBS-based SNPs. Five QTLs were detected on pepper genetic linkage map constructed by GBS-based SNPs and the developed HRM markers.

(11) line 207-208

“B genotype of SNP 208 C06_204307935 linked to the QTL Bwr6w-6.1 showed more resistance than A and H genotypes”. Why not check what gene is positioned nearby?

Authors’ reply> We are preparing for next manuscript for candidate genes involved in the regions mapped to the QTL.

(12) section 2.5

It will be better if the QTLs detected by different pathogen isolate named differently.

HRM markers. Please show the pictures of melting curves for important markers stressed by the authors.

Authors’ reply> We provided the pictures of melting curves for HRM markers linked to the QTLs (Figure S4)

(13) line 222

“The hybrid variety ‘Konesian Hot’ used for stock”, this information is important. Please stress the significance why choose this cultivar and provide more information like this in M&M, even in section of introduction.

Authors’ reply> Detailed information for ‘Konesian Hot’ was referred to Reference 4 article.

(14) line 234-line 240

What is the conclusion of this discussion. The QTLs identified in this study positioned at the same place with others? Different? How about comparing positions of QTLs between HS and HWA?

Authors’ reply> The QTLs on the same chromosome were discussed at page 13.

(15) line 367-368

Please show the picture of genotyping for important makers.

Authors’ reply> We added the picture of genotyping for HRM markers (Figure S4).

(16) table 2

Will the authors explain why use different sequencing platform?

Authors’ reply> We wanted to know the sequencing efficiency according to sequencing platforms, HiSeq 2500 and HiSeq X. HiSeq X was more efficient than HiSeq 2500.

(17) fig 2 and fig3

How about integrate QTL7.1 and QTL7.2 into one map?

Authors’ reply> Fig.2 and Fig.3 are different maps constructed in different F2 populations. The number of SNP marker indicates the position (bp) of physical map. Therefore, positions of two QTLs could be easily compared.

(18) figure 4

Is there statistically significance? I think some markers are not valuable.

Authors’ reply> It was statistically significant with Duncan’s multiple range test at 5% level (page 15).

(19) section 4.3

What is the culture condition of seedlings used to do resistance assay? Pot? Cell tray? Field? The resistance investigation was performed at the 6th week after inoculation? Really. It is too long in my knowledge. We do this experiment at 10th day.

Authors’ reply> We cultivated pepper seedlings in cell tray. We inoculated with inoculums of R. solanacearum at 6~8 leaf stage. We scored the disease index at the 6th week after inoculation.
